# Effects of Low Mental Energy from Long Periods of Work on Brain-Computer Interfaces

**DOI:** 10.3390/brainsci12091152

**Published:** 2022-08-29

**Authors:** Kaixuan Liu, Yang Yu, Ling-Li Zeng, Xinbin Liang, Yadong Liu, Xingxing Chu, Gai Lu, Zongtan Zhou

**Affiliations:** College of Intelligence Science and Technology, National University of Defense Technology, Changsha 410073, China

**Keywords:** BCI, electroencephalogram, low mental energy

## Abstract

Brain-computer interfaces (BCIs) provide novel hands-free interaction strategies. However, the performance of BCIs is affected by the user’s mental energy to some extent. In this study, we aimed to analyze the combined effects of decreased mental energy and lack of sleep on BCI performance and how to reduce these effects. We defined the low-mental-energy (LME) condition as a combined condition of decreased mental energy and lack of sleep. We used a long period of work (>=18 h) to induce the LME condition, and then P300- and SSVEP-based BCI tasks were conducted in LME or normal conditions. Ten subjects were recruited in this study. Each subject participated in the LME- and normal-condition experiments within one week. For the P300-based BCI, we used two decoding algorithms: stepwise linear discriminant (SWLDA) and least square regression (LSR). For the SSVEP-based BCI, we used two decoding algorithms: canonical correlation analysis (CCA) and filter bank canonical correlation analysis (FBCCA). Accuracy and information transfer rate (ITR) were used as performance metrics. The experimental results showed that for the P300-based BCI, the average accuracy was reduced by approximately 35% (with a SWLDA classifier) and approximately 40% (with a LSR classifier); the average ITR was reduced by approximately 6 bits/min (with a SWLDA classifier) and approximately 7 bits/min (with an LSR classifier). For the SSVEP-based BCI, the average accuracy was reduced by approximately 40% (with a CCA classifier) and approximately 40% (with a FBCCA classifier); the average ITR was reduced by approximately 20 bits/min (with a CCA classifier) and approximately 19 bits/min (with a FBCCA classifier). Additionally, the amplitude and signal-to-noise ratio of the evoked electroencephalogram signals were lower in the LME condition, while the degree of fatigue and the task load of each subject were higher. Further experiments suggested that increasing stimulus size, flash duration, and flash number could improve BCI performance in LME conditions to some extent. Our experiments showed that the LME condition reduced BCI performance, the effects of LME on BCI did not rely on specific BCI types and specific decoding algorithms, and optimizing BCI parameters (e.g., stimulus size) can reduce these effects.

## 1. Introduction

Electroencephalogram (EEG)-based brain–computer interfaces (BCIs) provide a new human-machine interaction strategy for users [1] and have been widely used in fields including disability assistance, education, and entertainment. BCI spellers are one of the most common BCI applications with which users can spell sentences and communicate with human companions [2,3]. In addition, BCI-controlled wheelchairs [4,5] and BCI-controlled robotic arms [6,7] are important in the BCI research field since they can be used in more complex tasks and more cluttered environments. For example, users can modulate the sensorimotor rhythms (SMRs)-based robotic arm developed by Hochberg et al. to bring them a bottle of water [6]. In [4], users simply needed to select a destination, and the BCI-controlled wheelchair introduced by Zhang et al. could automatically navigate to any location they selected. Although BCIs are useful in many scenarios, relevant studies suggested that the performance of BCIs might be affected by the user’s mental energy [8,9,10], which may hinder the applications of BCIs.

The term mental energy is a complex and interesting psychological concept, which is defined as a person’s ability to perceive the current environment and complete tasks [11]. Higher mental energy indicates that the brain is more active, and the person can more quickly and accurately perceive the environment and complete a prespecified task, and vice versa. A person’s mental energy is not constant; usually a long period of work will significantly reduce a person’s mental energy, and the longer a person works, the lower the mental energy [11,12,13]. For example, for the same person, the mental energy after working for several hours is lower than that when not working [14,15].

In our study, we defined the low-mental-energy (LME) condition as a combined condition of decreased mental energy and lack of sleep. Relevant studies suggest that reading books [16] and playing games [17,18] can evoke the user’s LME condition. For example, in a study conducted by Dimitrakopoulos et al. [17], Dimitrakopoulos et al. wanted to analyze the brain functional connectivity of users in the LME condition. To evoke the LME condition, Dimitrakopoulos et al. asked each user to play a driving game for several hours before participating in their experiments. In the present study, we used a similar method to evoke LME. To the best of our knowledge, few studies have examined the effects of LME on BCI performance. For this reason, we chose to assess these combined effects in this study.

P300-based BCIs [1] (BCIs that decode user commands from P300 signals) and SSVEP-based BCIs [19] (BCIs that decode user commands from steady-state visually evoked potential signals) have been widely used in various research fields. In our study, we chose to test the effects of LME on a P300-based BCI and a SSVEP-based BCI. The number of options of P300- and SSVEP-based BCIs in different scenarios might be different, but the EEG collection methods and the EEG-decoding algorithms used by BCIs are not severely affected by the number of options [19,20]. In our study, we set six options for the P300- and SSVEP-based BCIs, respectively. Each subject used the P300- and SSVEP-based BCIs to choose prespecified targets in normal conditions (as the control group) and LME conditions (as the experimental group). We analyzed the amplitude and signal-to-noise ratio (SNR) of the evoked EEG signals. Additionally, we measured the degree of fatigue and task load for each subject with the fatigue questionnaire introduced by Trudie et al. [21,22] and the NASA task load index (NASA-TLX) [23,24].

The original hypothesis of this study was that (a) LME can significantly reduce the BCI performance; (b) compared with the normal condition, the amplitude and SNR of the evoked EEG signals in the LME condition might be smaller, while the degree of fatigue and task load of each user in the LME condition might be larger; (c) in the LME condition, selecting appropriate stimulation parameters might improve the BCI performance to some extent.

The remainder of this paper is organized as follows: Section 2 describes the subjects, architecture of the BCIs, experimental procedure, EEG collection, signal-decoding algorithm, and performance metrics of the BCIs. Section 3 and Section 4 describe and discuss the experimental results. Section 5 concludes this paper.

## 2. Materials and Methods

A.Subjects

Ten subjects participated in this study, who ranged from 23 to 32 years old (4 females and 6 males) and had normal vision or normal vision after correction. According to Lieberman et al. [11], taking drugs, and drinking stimulating beverages such as coffee could change a person’s mental energy, which might interfere with our study. Before the experiment, we used self-reported measurements. People who had a disease, had a sleep disorder, or drank coffee 24 h before the experiment were excluded. Before the experiments, the subjects should have kept enough sleep: the sleeping number of hours was 8.5 ± 1.0. This study was approved by the Institutional Review Board of Xiangya Hospital of Central South University and was conducted in accordance with the ethical standards of the 1964 Declaration of Helsinki. All subjects provided written consent before the experiment. The experiment was carried out in a normal room that was 5.0 m × 3.0 m in size. During the experiment, all doors and windows were closed, and only the subjects and researchers were allowed to stay in the room. Four 1.5 m-long 18-watt incandescent lamps were used for lighting. The subjects sat in a chair 60 cm from the laptop screen (17.3 inches, 1920 × 1000 pixels, 144 Hz). Before the experiment, we described the procedure and purpose of the experiment to each subject.

B.Architecture of the BCIs

Figure 1 shows the architecture of the P300-based BCI and the SSVEP-based BCI we used, both of which consisted of two modules: the user interface module and the signal decoding module.

The user interface of the P300- and SSVEP-based BCI was a laptop screen (Hasee Super Ares, Hasee Computer, Shenzhen, China), which showed the BCI options, the stimuli, the decoding results, and a visual cue used to remind subjects of the current target. In our study, the user interface included six options: “Box”, “Bottle”, “Chair”, “Book”, “Laptop”, and “Phone”. As shown in Figure 2, the time courses of the P300- and SSVEP-based BCI included three stages: cue, flash, and decode. For the P300-based BCI, in the cue stage, our system automatically marked the current target option with a blue visual cue. Based on Hick’s law, the cue stage should last for one second [25]. In the flash stage, our system used blue rectangular stimuli to evoke the user’s P300 signals. The stimuli flashed in accordance with the single-character (SC) paradigm [26,27]. The stimulus onset asynchrony (SOA) of each stimulus was 400 ms (i.e., each rectangle appeared for 200 ms and disappeared for 200 ms). Each stimulus flashed four times [28]. In the decoding stage, our P300-based BCI would show the decoding results on the user interface. For the SSVEP-based BCI, our system also marked the current target option in the cue stage. In the flash stage, each option on the user interface flashed at different frequencies: the options “Box”, “Bottle”, “Chair”, “Book”, “Laptop”, and “Phone” flashed at frequencies of 6.2, 7.1, 8.0, 8.9, 9.8, and 11.0 Hz, respectively [20]. The flash stage of our SSVEP-based BCI lasted for 3000 ms [29]. In the decoding stage, our SSVEP-based BCI would feed the decoding results back to the users. Although we asked each subject to focus on the current target during the experiment, sometimes a subject might watch non-target positions because of a lack of interest in the experiment. For example, during the experiment, we found that subject S1 was looking at the devices near the laptop instead of the targets shown in the user interface seven times. Relevant studies have shown that if users do not focus on the current target, BCI performance will decrease [30,31]. In this study, we wanted to determine the effects of the LME condition on BCI performance. Therefore, we excluded trials involving this type of irrelevant interference. In our study, we used a Tobii eye tracker (Tobii 4 C, Tobii Tech, Danderyd, Sweden) to track each subject’s point of gaze (POG). When a subject’s POG left the current target, we did not record the EEG signals for this target.

The signal-decoding module consisted of two submodules. The first was an amplifier and its Ag/AgCl electrodes (actiCHamp, Brain Products, Gilching, Germany), which read, amplified, and preprocessed the raw EEG signals from the subject’s scalp. The second was a laptop-run BCI2000 platform, which decoded user commands from the preprocessed EEG signals [32]. In this study, the subjects could switch between the two BCIs with a small switch: if subjects pressed the switch once, the P300-based BCI would be activated; if the switch was pressed again, the P300-based BCI would be turned off and the SSVEP-based BCI would be activated.

P300 decoding algorithm: Every time a stimulus appeared on an option, we started recording EEG signals. We recorded a 700 ms-long EEG signal for each stimulus. The sampling frequency we used was 200 Hz. Therefore, for each stimulus, we collected a feature vector that contained 140 points. Next, we used a down-sampling filter with a sampling rate of one tenth to process each feature vector. The purpose of this step was to reduce the data size. After this step, each feature vector contained 14 points. Next, we used two widely used EEG feature translation methods, i.e., the stepwise linear discriminant analysis (SWLDA) algorithm [33] and least square regression (LSR) [34], to decode each feature vector. We used the EEG data collected in the BCI calibration phase to train the optimal weight matrix W required by the SWLDA and LSR decoding algorithms. In the decoding process, we multiplied the optimal weight matrix W by each feature vector, and the product was the score for each stimulus. We obtained the score of each option by adding the scores of all stimuli that appeared on this option. Finally, we used the bubble sort algorithm to determine the largest score. The option with the largest score was the decoding result [35].
(1)$${\mathrm{Score}}_{i}=\sum_{j=1}^{k}W^{T}X_{\mathrm{ij}}$$

In Equation (1), W is the optimal parameter matrix obtained in BCI calibration and k is the number of flashes. $X_{\mathrm{ij}}$ is the signal of the $i^{\mathrm{th}}$ option for the $j^{\mathrm{th}}$ stimulus. ${\mathrm{Score}}_{i}$ is the score of the $i^{\mathrm{th}}$ option.

SSVEP decoding algorithm: The flash duration of the SSVEP-based BCI was 3000 ms, so we collected a 3000 ms-long EEG signal before decoding a user’s command. The sampling frequency we used was 200 Hz. We used a Chebyshev type I filter (6th order, the passband was [4 Hz, 35 Hz], ripple in the passband was 0.1 dB) to process the collected EEG signal. Next, we used two widely used SSVEP feature translation methods, i.e., the canonical correlation analysis (CCA) algorithm [36] and filter bank canonical correlation analysis (FBCCA) algorithm [37], to decode the user’s commands. In the decoding process, the CCA algorithm would calculate the correlation coefficients between the EEG signal $Y_{\mathrm{EEG}}$ and the flash frequency signal of each option $Y_{\mathrm{opt}}^{i}$. The SSVEP-based BCI had six options, and each option had a unique flash frequency. Therefore, we could obtain six flash frequency signals $Y_{\mathrm{opt}}^{i}$ and six correlation coefficients $r_{i}$. We used the bubble sort algorithm to determine the option with the largest correlation coefficient, which was the decoding result. In Equations (2) and (3), i represents the $i^{\mathrm{th}}$ option, $f_{i}$ and $r_{i}$ are the flash frequency and the correlation coefficient of the $i^{\mathrm{th}}$ option, respectively, and t is the sampling time point. Coor is the correlation coefficient operator.
(2)$$Y_{\mathrm{opt}}^{i}=\left[\sin\left(2{\pi f}_{i}t\right),\cos\left(2{\pi f}_{i}t\right),\ldots ,\sin\left(6{\pi f}_{i}t\right),\cos\left(6{\pi f}_{i}t\right)\right]$$
(3)$$r_{i}=\mathrm{Coor}\left(Y_{\mathrm{opt}}^{i},Y_{\mathrm{EEG}}\right)$$

The decoding process of the FBCCA algorithm included the below steps: Firstly, we used several bandpass filters to perform sub-band decomposition for each 3000 ms-long EEG signal $Y_{\mathrm{EEG}}$, and the passbands of the used bandpass filters are shown in Equation (4). According to the Nyquist sampling theorem, the minimum sampling frequency required is 92 Hz < 200 Hz. Therefore, we can record the SSVEP signal at a sampling frequency of 200 Hz and decode the SSVEP signal by the FBCCA algorithm.

Next, the standard CCA process was applied to each sub-band component separately, i.e., we calculated the correlation coefficients between each sub-band ${\mathrm{SB}}_{k}\;\left(k=1,2,3\ldots ,7\right)$ and each flash frequency signal $Y_{\mathrm{opt}}^{i}\;\left(i=1,2,3,..,6\right)$. As shown in Equation (5), vector $r_{i}{}^{\mathrm{vec}}$ was the calculated correlation coefficients. Next, as shown in Equations (6) and (7), we used the coefficients $a\;\left(n\right)$ [37] to determine the weighted sum of $a\;\left(n\right)$ and $r_{i}{}^{\mathrm{vec}}$. We used the bubble sort algorithm to determine the option with the largest $r_{i}$, which was the decoding result.
(4)$$\left\{\begin{array}{c} {\mathrm{SB}}_{1}=\left[6\;\mathrm{Hz},\;11\;\mathrm{Hz}\right]\;\\ {\mathrm{SB}}_{2}=\left[11\;Hz,\;16\;Hz\right]\\ {\mathrm{SB}}_{3}=\left[16\;Hz,\;21\;Hz\right]\\ {\mathrm{SB}}_{4}=\left[21\;Hz,\;26\;Hz\right]\\ {\mathrm{SB}}_{5}=\left[26\;\mathrm{Hz},\;31\;\mathrm{Hz}\right]\\ {\mathrm{SB}}_{6}=\left[31\;\mathrm{Hz},\;36\;\mathrm{Hz}\right]\\ {\mathrm{SB}}_{7}=\left[41\;Hz,\;46\;Hz\right] \end{array}\right.$$
(5)$$r_{i}{}^{\mathrm{vec}}=\left[\begin{array}{c} r_{i}{}^{1}\\ r_{i}{}^{2}\\ r_{i}{}^{3}\\ r_{i}{}^{4}\\ r_{i}{}^{5}\\ r_{i}{}^{6}\\ r_{i}{}^{7} \end{array}\right]=\left[\begin{array}{c} \begin{array}{c} Coor\left(Y_{\mathrm{opt}}^{i},{\mathrm{SB}}_{1}\right)\\ Coor\left(Y_{\mathrm{opt}}^{i},{\mathrm{SB}}_{2}\right)\\ Coor\left(Y_{\mathrm{opt}}^{i},{\mathrm{SB}}_{3}\right)\\ Coor\left(Y_{\mathrm{opt}}^{i},{\mathrm{SB}}_{4}\right)\\ Coor\left(Y_{\mathrm{opt}}^{i},{\mathrm{SB}}_{5}\right)\\ Coor\left(Y_{\mathrm{opt}}^{i},{\mathrm{SB}}_{6}\right) \end{array}\\ Coor\left(Y_{\mathrm{opt}}^{i},{\mathrm{SB}}_{7}\right) \end{array}\right]$$
(6)$$a\left(n\right)=n^{-1.25}+0.25,\;n\in \left[1,2,3,\ldots ,7\right]$$
(7)$$r_{i}=\sum_{n=1}^{7}a\left(n\right)*{\left(r_{i}{}^{n}\right)}^{2}$$

C.Experimental Procedure

Each subject participated in two experiments within a week. The two experiments were assigned in a random order to avoid order effects. To prevent the subjects from remembering or being too familiar with the experimental paradigms, the subjects could not participate in the two experiments on the same day. In our study, the interval between the two experiments was equal to or greater than three days. We wanted to give each subject enough time to relax and recover from the first experiment, so as not to interfere with the second experiment. One experiment was conducted in normal conditions and was defined as the normal condition experiment. The other experiment was conducted when the subjects were in an LME condition, which was defined as the LME-condition experiment. In daily life, typical working scenarios involve reading academic books and playing games, both of which could effectively evoke LME [17]. We prepared some books in advance based on each subject’s job and hobbies. The game we used was CARLA Simulator (Version 0.9.9.4, http://carla.org/, accessed on 22 April 2020) [38]. Relevant studies have suggested that short durations of work could not well reduce mental energy and that subjects might recover during the course of the experiment [39]. Working too long would have required too much time and might not be feasible, and 18 h was a good choice [14].

Figure 3 illustrated our experiment design. In our study, each subject entered our laboratory at approximately 8:00 a.m. Before the LME-condition experiment, each subject had to read the books we prepared and play the driving game. Each subject could randomly choose to play games or read books, but the total time spent reading books and playing the game must be 18 h. For example, a subject could first read books for 3 h and then play games for the remaining 15 h. During the 18 h, each subject was not allowed to sleep or go out. If a subject was too tired to continue, the subject was allowed to take a few minutes of rest, e.g., walking in our laboratory. In order to better control the reading activities, we would randomly ask questions about the contents each subject was reading. We asked once an hour. If the reading time was less than one hour, it would be regarded as one hour. For example, if a subject read for 9.5 h, we would ask questions ten times. In order to better control the driving activities, we set up two cars in the driving game, one driven automatically by the computer and the other driven by the subjects. The subjects were asked to follow the self-driving car (the distance should be maintained between 10 and 15 m) and avoid obstacles such as pedestrians. At around 2:00 a.m. the next day, the LME experiment began. Before the normal-condition experiment, each subject was not allowed to work, and each subject was asked to rest but not sleep for one hour in our laboratory, e.g., subjects could listen to some music or just sit on a comfortable chair. At around 9:00 a.m. the same day, the normal experiment began.

In our study, each subject participated in two experiments: the normal-condition experiment and the LME-condition experiment. The procedures of the two experiments were the same, and both contained three phases: (1) the BCI calibration phase, (2) the online phase, and (3) the optimization phase. In the BCI calibration phase, each subject used the P300-based BCI to select 20 predefined targets. In this phase, we collected the EEG signals to train the SWLDA and LSR decoding algorithms, and obtained the optimal weight matrix. In the online phase of the normal-condition experiment and the LME-condition experiment, each subject used the P300-based BCI and the SSVEP-based BCI to choose 30 prespecified targets, respectively. Each subject could first use the P300-based BCI and then use the SSVEP-based BCI, or use the two BCIs in a reverse order. The P300 signal is mainly observed near the parietal lobe and the SSVEP signal is mainly observed near the occipital lobe. In our study, the average P300 signal is defined as the average signal of the P300 signals collected by five electrodes (FC1, FC2, CP1, CP2, and Cz). The average SSVEP signal is defined as the average signal of the SSVEP signals collected by six electrodes (PO3, O1, POZ, Oz, O2, and PO4). Figure 4 showed the EEG electrodes used in our study. We created Figure 5 below, Table 1 and Table 2 using the data collected in the online phase of the two experiments. Additionally, according to the below Equations (10)–(12), we calculated the amplitudes and SNRs of the EEG signals recorded in the online phase of the two experiments, and the results are shown in Figure 6. In the optimization phase of the normal-condition experiment and the LME-condition experiment, each subject completed the following two additional tasks using the P300- and SSVEP-based BCIs. Figure 7 was created using the data collected in the optimization phase of the two experiments.

Task (a): in this task, we resized stimuli of the P300- and SSVEP-based BCIs in the following order: 100 × 100 pixels, 150 × 150 pixels, 200 × 200 pixels, 250 × 250 pixels, and 300 × 300 pixels. For the P300-based BCI, each subject chose 10 targets at each stimulus size. Specifically, each subject first chose 10 targets with the 100 × 100 pixel stimuli, then chose 10 targets with the 150 × 150 pixel stimuli, and so forth, until the subject completed the task. In other words, each subject chose a total of 50 targets with the P300-based BCI. For the SSVEP-based BCI, each subject also chose 10 targets at each stimulus size. In task (a), the number of flashes of the P300-based BCI remained at four, and the flash duration of the SSVEP-based BCI remained at 3000 ms.

Task (b): in this task, we changed the number of flashes of the P300-based BCI (two, three, four, five, and six) and the flash duration of the SSVEP-based BCI (2000 ms, 3000 ms, 4000 ms, 5000 ms, and 6000 ms). The stimuli size of the two BCIs remained 100 × 100 pixels. For the P300-based BCI, each subject chose 10 targets with two flashes, and then chose 10 targets with three flashes, and so forth, until six flashes. For the SSVEP-based BCI, each subject chose 10 targets with the flash duration of 2000 ms, and then chose 10 targets with the flash duration of 3000 ms, and so forth, until a flash duration of 6000 ms.

D.Task Load and Fatigue Degree

The degree of fatigue and the task load might affect the BCI performances. In our study, we used the NASA-TLX to measure the task load for each subject. The NASA-TLX [23,24] included six items: “mental demand”, “physical demand”, “temporal demands”, “frustration”, “effort”, and “performance”. The full score for each item was 100, and subjects separately rated each item. A high score indicated that the load for that particular item was high, and conversely, a low score indicated low load. For example, if the score of the item “effort” was 100, it meant that the subject made a large effort to complete the current task. We used the fatigue questionnaire introduced by Trudie et al. to measure each subject’s degree of fatigue. The fatigue questionnaire [21,22] included 14 items: seven items measured the physical fatigue of a person and the other seven items measured the mental fatigue. The full score of the fatigue questionnaire was 100, and each item was worth approximately seven points. The higher the scores were, the more fatigued the person. We measured each subject’s task load and degree of fatigue immediately after the experiments and present the scores in Table 3 to Table 4 (see Results Section 3).

E.EEG Collection and Preprocessing

The EEG signals were collected from 11 electrodes—FC1, FC2, CP1, CP2, Cz, PO3, O1, POz, Oz, O2, and PO4—distributed according to the international 10–20 system and amplified with an actiCHamp amplifier (Brain Products, Gilching, Germany). The sampling frequency was 200 Hz. TP9 and TP10 were reference electrodes, and the reference corresponds to the average of the signals from these two electrodes. Fpz was the ground electrode. The impedance of all electrodes was maintained below 5 kΩ. To reduce artifacts related to eye and muscle activities on the EEG signal, we filtered the collected EEG signal using a bandpass filter from 0.5 to 50 Hz.

F.Metrics

(i)BCI performance (accuracy and ITR): Relevant studies often use the ITR and accuracy to evaluate the performance of BCIs [19]. In this study, we also used the ITR and accuracy as performance metrics. Equations (8) and (9) show the method to compute the ITR. N is the number of options, which was set to 6, p is the accuracy, and T is the time required for decoding a target.
(8)$$\mathrm{ITR}=\frac{\log_{2}N+{p*\log}_{2}p+\left(1-p\right){*\log}_{2}\left(\frac{1-p}{N-1}\right)}{T},\;\mathrm{if}\;p\neq 100\%$$
(9)$$\mathrm{ITR}=\begin{array}{cc} \frac{\log_{2}N}{T}, & \mathrm{if}\;p=100\% \end{array}$$(ii)EEG indices (amplitude and SNR): Studies have suggested that the amplitude and SNR of evoked EEG signals are associated with the performance of BCIs [20,40]. According to [27,41], the SNR of a P300 signal can be computed with Equations (10) and (11). In the two equations, k denotes the $k^{\mathrm{th}}$ flash. ${\mathrm{EEG}}_{k}$, $P300_{k}$, and ${\mathrm{Noise}}_{k}$ are the recorded EEG signal in the $k^{\mathrm{th}}$ flash, the true P300 signal in the $k^{\mathrm{th}}$ flash, and the uncorrelated noise components in the $k^{\mathrm{th}}$ flash, respectively. $E_{k}$ denotes the mathematical expectation operator, namely, the computed mean signal of all flashes. The mean of the noise component ${\mathrm{Noise}}_{k}$ was assumed to be zero, and $E_{k}\left({\mathrm{EEG}}_{k}\right)$ returned the true P300 signal. ${\mathrm{var}}_{t}$ computed the variance in the signals over time.
(10)$${\mathrm{EEG}}_{k}=P300_{k}+{\mathrm{Noise}}_{k}$$
(11)$$\mathrm{SNR}=10\log_{10}\left(\frac{{\mathrm{var}}_{t}\left(E_{k}\left({\mathrm{EEG}}_{k}\right)\right)}{E_{k}\left({\mathrm{var}}_{t}\left({\mathrm{EEG}}_{k}-E_{k}\left({\mathrm{EEG}}_{k}\right)\right)\right)}\right)$$

According to [20], the SNR of an SSVEP signal was the power at the flash frequency divided by the mean power at its neighboring frequency band. If the flash frequency was $f_{i}$ Hz, its neighboring frequency band was defined as $\left[f_{i}-2\mathrm{Hz},{\;f}_{i}+2\mathrm{Hz}\right]$ [20]. For the SSVEP-based BCI we used, the frequency resolution was approximately 0.5 Hz. Therefore, $\left[f_{i}-2\mathrm{Hz},{\;f}_{i}+2\mathrm{Hz}\right]$ contained 10 points, i.e., from 5 points on the left side of $f_{i}$ Hz to 5 points on the right side. Equation (12) shows the SNR of a SSVEP signal. $P\left(f_{i}\right)$ and ${\mathrm{SNR}}_{i}$ are the power and the SNR at $f_{i}$ Hz, respectively.
(12)$${\mathrm{SNR}}_{i}=10\log_{10}\left(P\left(f_{i}\right)/\left(\frac{1}{10}\sum_{j=-5,j\neq 0}^{j=5}P\left(f_{i+j}\right)\right)\right)$$

G.Statistical Analysis

The data we recorded (accuracies and ITRs) followed normal distributions, and the statistical power of the paired *t* test was higher than that of non-parametric test. Therefore, in this study, we used a paired *t* test to compare the BCI performance in normal and LME conditions. The statistical analysis software we used was the SPSS (version 21.0). For example, in the following Table 1, the first item is “Accuracy by SWLDA decoding”. For this item, we compared the accuracy in the normal condition with the accuracy in the LME condition. We collected the data of each subject under different stimulus parameters (flash duration, flash number, and stimulus size), which is a repeated-measure design. Additionally, the data we collected followed normal distributions, and the statistical power of analysis of variance (ANOVA) was higher than that of the non-parametric test. Therefore, in our study, we used one-way repeated ANOVA to test the effects of stimulus parameters.

## 3. Results

A.Performance of the BCIs

Table 1 shows the performance of the P300-based BCI. The mean accuracy in the LME condition was 46.4% (decoded by the SWLDA classifier) and 43.6% (decoded by the LSR classifier), respectively; the mean ITR was 1.8 bits/min (decoded by the SWLDA classifier) and 1.6 bits/min (decoded by the LSR classifier), respectively. In the normal condition, the mean accuracy was 81.3% (decoded by the SWLDA classifier) and 83.4% (decoded by the LSR classifier), respectively; the mean ITR was 7.8 bits/min (decoded by the SWLDA classifier) and 8.3 bits/min (decoded by the LSR classifier), respectively. Table 2 shows the performance of the SSVEP-based BCI. In the LME condition, the mean accuracy was 48.7% (decoded by the CCA classifier) and 53.7% (decoded by the FBCCA classifier), respectively; the mean ITR was 5.1 bits/min (decoded by the CCA classifier) and 6.4 bits/min (decoded by the FBCCA classifier), respectively. In the normal condition, the mean ITR accuracy was 91.0% (decoded by the CCA classifier) and 93.4% (decoded by the FBCCA classifier), respectively; and the mean ITR was 23.6 bits/min (decoded by the CCA classifier) and 25.2 bits/min (decoded by the FBCCA classifier), respectively. The results of Shapiro-Wilk tests [42] showed that all accuracies (*p* > 0.01) and ITRs (*p* > 0.01) in Table 1 and Table 2 followed normal distributions. Next, we used paired *t* tests to further analyze the effects of LME on performance of the P300- and SSVEP-based BCIs. The experimental results showed that in the LME conditions, the performance of P300-based BCI (using the SWLDA classifier or the LSR classifier) and the SSVEP-based BCI (using the CCA classifier or the FBCCA classifier) was significantly worse (*p* < 0.001).

B.EEG Signals in Normal and LME Conditions

In the normal condition, an obvious P300 response can be observed in Figure 5A, and the amplitude of the P300 signal was approximately 2.0 μV. In the LME condition, we could also see a P300 response (approximately 1.0 μV) at approximately 400 ms, but the P300 response was not as clear as that in the normal condition. With the SSVEP-based BCI, each stimulus evoked obvious base frequencies and harmonic frequencies. In the normal condition, the maximum amplitude of base frequencies was approximately 3 μV, which was observed at 8 Hz, and the amplitudes of other base frequencies were approximately 2 μV. In the LME condition, the maximum amplitude of the base frequencies was approximately 1.5 μV, and the other amplitudes were approximately 1 μV, which were less than those recorded in the normal condition. In the LME condition, the amplitudes of all harmonic frequencies were also less than those recorded in the normal condition. LME decreased the amplitudes of the evoked SSVEP signals.

We calculated the amplitudes and SNRs of the average P300 signals and the average SSVEP signals, and showed the results in Figure 6. The mean amplitudes in the LME and normal conditions were 1.2 V and 2.2 V, respectively. The mean SNRs in the LME and normal conditions were 4.5 dB and 10.2 dB, respectively. The results from the Shapiro-Wilk tests showed that all amplitudes (*p* > 0.01) and SNRs (*p* > 0.01) followed normal distributions. Next, we used paired *t* tests. The amplitudes and SNRs in the LME condition were smaller than those in the normal condition (*p* < 0.001). Generally, if the amplitudes and SNRs of the evoked EEG signals are larger, it indicates that the BCI paradigms successfully evoked higher-quality EEG signals [20]. Figure 6 suggests that the quality of the EEG signals with the P300-based BCI and SSVEP-based BCI were not as high in the LME condition as those in the normal condition.

C.NASA-TLX and Fatigue Questionnaire Scores

Table 3 and Table 4 show the overall scores from the NASA-TLX and the fatigue questionnaire, respectively. Despite some individual differences, in the LME condition, the mean scores of mental demand, physical demand, temporal demands, frustration, effort, and performance were all approximately 80. In the normal condition, the mean scores of these metrics were all approximately 30. In the LME condition, the mean scores of physical fatigue and mental fatigue were 85.2 and 62.0, respectively. In the normal condition, the two metrics were 20.4 and 32.1, respectively. Paired *t* tests have often been used to test the results from the NASA-TLX and the fatigue questionnaire [43], and we also used paired *t* tests in our study. All scores of the NASA-TLX and the fatigue questionnaire in the LME condition were significantly higher than those in the normal condition (*p* < 0.001). Regarding the NASA-TLX and the fatigue questionnaire, higher scores indicated that the tasks were more difficult for the subjects, and the subjects were more fatigued at their completion [21,22,23,24]. The tasks in the LME and normal conditions were the same. The results might indicate that in the LME condition, the subject’s cognitive ability was reduced and they were easier to be fatigued.

D.Effects of Flash Duration, Flash Number, and Stimulus Size

To test our hypothesis (c), in the optimization phase of the experiment, we changed three parameters of the BCIs: flash duration, flash number, and stimulus size. In Figure 7, we show the accuracies and ITRs of the P300- and SSVEP-based BCIs using the different parameters. Although the accuracies and ITRs of the two BCIs in the LME condition were lower than those in the normal condition, increasing the three parameters improved the accuracies and ITRs of the BCIs in the LME condition to some extent. To further validate the experimental results, we used one-way repeated ANOVA to test the data shown in Figure 7. For the accuracy of the P300-based BCI, the results showed that increasing the flash number (for LSR decoding: F(4,36) = 3.7, *p* = 0.01; for SWLDA decoding: F(4,36) = 2.5, *p* = 0.05) and stimulus size (for LSR decoding: F(4,36) = 6.0, *p* = 0.001; for SWLDA decoding: F(4,36) = 9.0, *p* = 0.000) could improve the BCI performance. For the ITR of the P300-based BCI, the results showed that increasing the flash number (for LSR decoding: F(4,36) = 66.0, *p* = 0.000; for SWLDA decoding: F(4,36) = 120.0, *p* = 0.000) and stimulus size (for LSR decoding: F(4,36) = 37.0, *p* = 0.000; for SWLDA decoding: F(4,36) = 3.8, *p* = 0.01) could improve the BCI performance. For the accuracy of the SSVEP-based BCI, the results showed that increasing the flash duration (for FBCCA decoding: F(4,36) = 19.0, *p* = 0.000; for CCA decoding: F(4,36) = 7.1, *p* = 0.000) and stimulus size (for FBCCA decoding: F(4,36) = 9.4, *p* = 0.000; for CCA decoding: F(4,36) = 9.8, *p* = 0.000) could improve the BCI performance. For the ITR of the SSVEP-based BCI, the results showed that increasing the flash duration (for FBCCA decoding: F(4,36) = 145.0, *p* = 0.000; for CCA decoding: F(4,36) = 8.5, *p* = 0.000) and stimulus size (for FBCCA decoding: F(4,36) = 14.0, *p* = 0.000; for CCA decoding: F(4,36) = 22.6, *p* = 0.000) could improve the BCI performance.

## 4. Discussion

With the P300-based BCI and the SSVEP-based BCI, LME reduced accuracies and ITRs. Although subjects completed the same tasks in the normal condition and the LME condition, the results from the NASA-TLX and the fatigue questionnaire showed that the subjects felt that the tasks were more difficult in the LME condition. In the LME condition, the subject might lack enough cognitive ability and enthusiasm for the tasks. The mental fatigue, frustration, and distraction caused by LME might be a reason for the decline in BCI performance. Our experimental results showed that the amplitudes and SNRs of the EEG signals evoked in the LME condition were significantly lower than those in the normal condition, which might be another reason for the decrease in BCI performance.

Increasing the number of flashes and flash duration provided more EEG data to the classifiers, which makes the decoding process easier [44]. Therefore, increasing the number of flashes and flash duration improved the performance of the P300- and SSVEP-based BCIs to some extent. In the LME condition, the users could not easily find small stimuli because their reaction speeds and attention were reduced. However, users can easily find large stimuli because large stimuli can better attract users’ attention [45,46,47,48]. The BCIs used in our study were visually evoked. The size of visual stimuli might influence visual-information-processing abilities in the human brain [49,50]. A relevant study showed that specific cells in the inferior temporal cortex responded only to a specific stimulus size [51]. A large visual stimulus might improve visual-information-processing abilities since it might activate more visual neurons [49], which might improve BCI performance in the LME condition to some extent.

At present, most studies on mental energy have tested the impact of cognitive workload on BCI performance. However, the decreased mental energy caused by high cognitive workload is different from that caused by long periods of work. The two types of decreases in mental energy have completely different neurophysiological mechanisms. Studies on the former could not adequately account for the effects of the latter on BCI performance [14,52]. Determining the effects of LME conditions on BCIs was the basis to optimize BCI interaction strategies in the LME condition. A BCI that could perform robustly in LME conditions would be useful for more scenarios and more potential users. For example, a disabled person may want to use BCIs after long periods of rehabilitation training, or a user may want to use BCIs after a full day of work. Additionally, potential users of BCI-controlled wheelchairs, such as patients with spinal injury, often continuously use BCI-controlled wheelchairs for several hours because they have to rely on wheelchairs in their daily life [53]. Although many studies have optimized BCI paradigms, users’ mental energy will be reduced after using these BCIs for long periods of time [26,54,55]. Even when not using BCIs, the mental energy of many patients is low, and using BCIs for long periods of time will more quickly reduce their mental energy. For the above scenarios, although the method of increasing stimulus size did not fully ensure BCI performance, it was useful to some extent. Moreover, combining our work with EEG-based fatigue detection may provide a more user-friendly interaction framework: the system automatically detects the user’s degree of fatigue from the recorded EEG signals and then adaptively chooses appropriate BCI parameters. We tested only the P300-based BCI and the SSVEP-based BCI, but this study, as a case study, can be extended to other types of BCIs.

Our research evaluating LME remains preliminary. We may integrate functional magnetic resonance imaging (fMRI) into our future work to determine more specific reasons why LME affects BCI performance. For example, in LME conditions, we could determine the relationship between brain functional connectivity and the SNR of EEG signals recorded in each brain region. We also need to develop a better BCI paradigm to further improve its performance in LME conditions. Laplacian spatial filters can filter some background noise and increase the SNRs of the recorded EEG signals, which are usually used to process sensorimotor rhythms [56,57]. It may be an interesting future direction to analyze P300 and SSVEP data with a Laplacian spatial filter. In our study, we did not collect enough data for two-way ANOVA, and it may be difficult to analyze the relation among flash number for P300, flash duration for SSVEP, and stimulus size, which is a limitation of our current work. Our study did not test the circadian/wake effects, which is a limitation. The lack of sleep may reduce people’s reaction speed and vigilance [58], but may increase the power of theta and beta waves. Sleep is also related to people’s moods, e.g., depression [59].

## 5. Conclusions

In this study, we examined the effects of LME due to a long period of work on BCI performance. The experimental results suggested that LME significantly reduced performances of the P300-based BCI and the SSVEP-based BCI. LME reduced user interest and enthusiasm for BCIs and caused more mental fatigue, frustration, and distraction. In addition, the amplitudes and SNRs of the evoked EEG signals were lower in the LME condition. We felt that these might be possible reasons for decreased BCI performance. Further results suggested that increasing the stimulus size, the flash number, and the flash duration could improve BCI performance in LME conditions to some extent.

## Figures and Tables

**Figure 1 brainsci-12-01152-f001:**
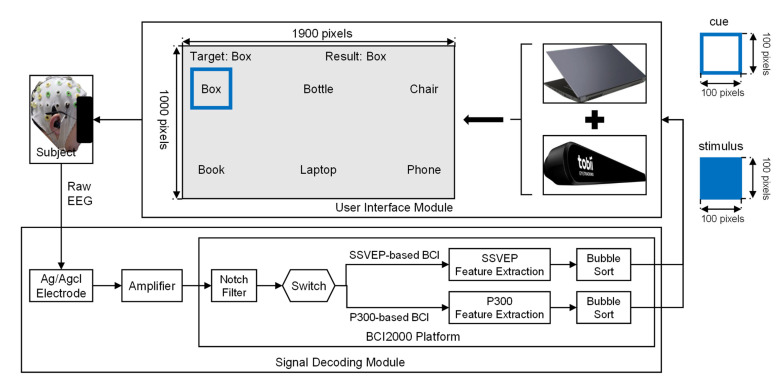
The architecture of the P300-based BCI and the SSVEP-based BCI used in this study. The two BCIs ran on the same laptop and could be converted between each other with a switch. The user interface of the two BCIs was a laptop screen with a size of 1900 pixels 1000 pixels; the size of each stimulus in the user interface was 100 pixels. We used a blue visual cue to mark the current target and a blue stimulus to evoke users’ EEG signals. The text after “Target” and “Result” presented the current target and decoding result, respectively.

**Figure 2 brainsci-12-01152-f002:**
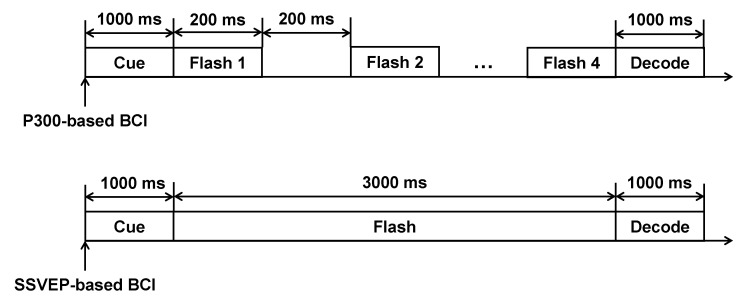
Illustration of the time courses.

**Figure 3 brainsci-12-01152-f003:**
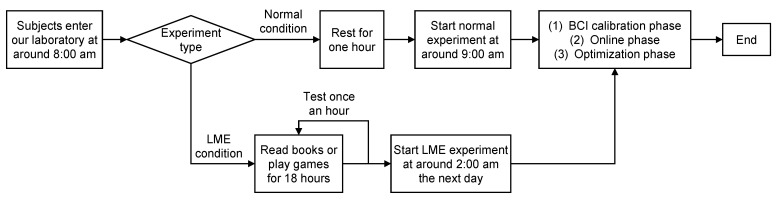
Illustration of our experiment design.

**Figure 4 brainsci-12-01152-f004:**
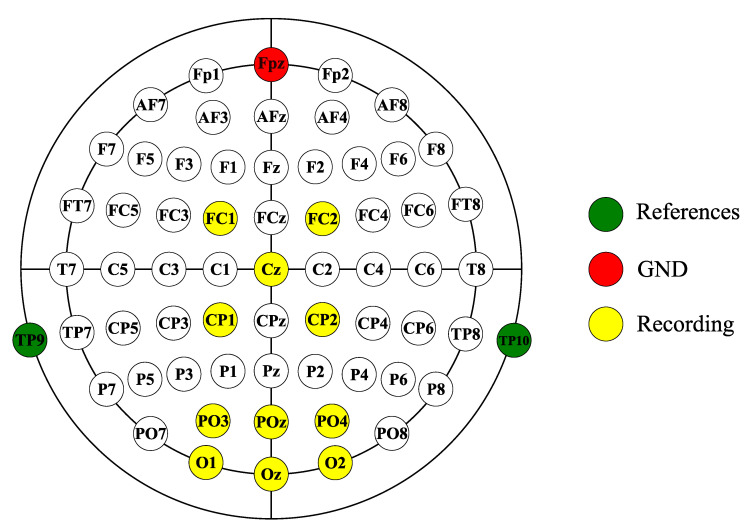
The EEG electrodes used in our study.

**Figure 5 brainsci-12-01152-f005:**
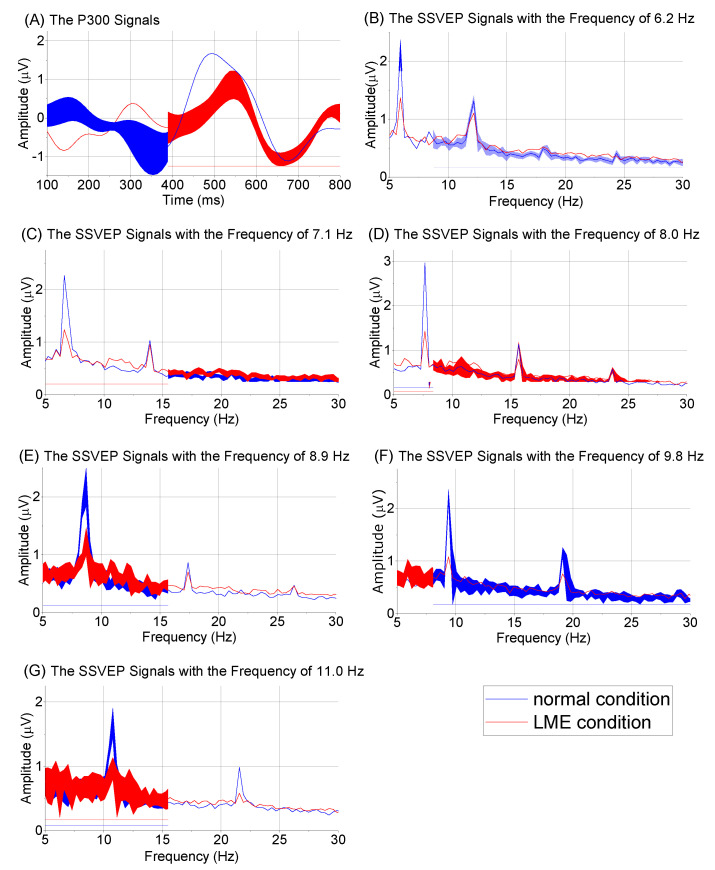
Evoked P300 signals and SSVEP signals.

**Figure 6 brainsci-12-01152-f006:**
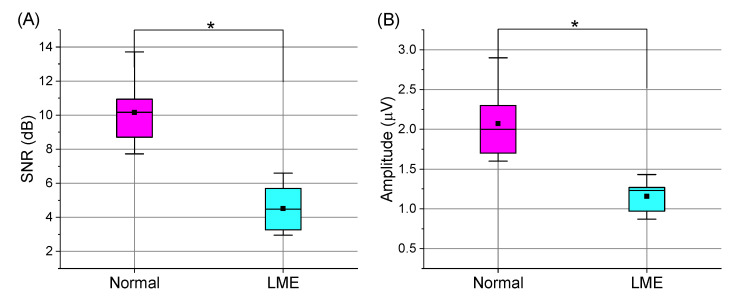
Amplitudes and SNRs in the Normal and LME Conditions. We plotted this figure using the EEG signals collected in the online phase of the normal and LME condition experiments. The subfigures (**A**,**B**) showed the amplitudes and SNRs of the collected EEG signals, respectively. In this figure, “*” means that the results were significantly different (*p* < 0.001).

**Figure 7 brainsci-12-01152-f007:**
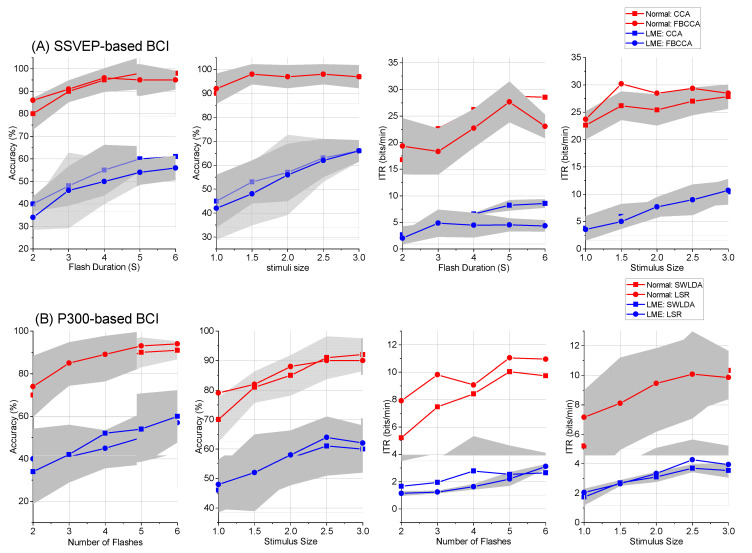
Effects of the flash number, flash duration, and stimulus size on BCI performance. The shaded areas in this figure represent the error bands. We plotted this figure using the EEG signals collected in the optimization phase of the normal and LME condition experiments. The subfigures (**A**,**B**) showed the experimental results of the P300- and SSVEP-based BCIs, respectively.

**Table 1 brainsci-12-01152-t001:** Accuracy and ITR with the P300-based BCI.

Subjects	Sex	Accuracy (%) by SWLDA Decoding		ITR (Bits/min) by SWLDA Decoding		Accuracy (%) by LSR Decoding		ITR (Bits/min) by LSR Decoding	
		Normal	LME	Normal	LME	Normal	LME	Normal	LME
S1	Female	83.3	46.7	8.0	1.8	80.0	33.3	7.2	0.6
S2	Female	100.0	40.0	13.4	1.1	83.3	43.3	8.0	1.5
S3	Male	76.7	43.3	6.5	1.5	90.3	46.7	9.8	1.8
S4	Female	80.0	56.7	7.2	3.1	93.0	57.0	10.6	3.1
S5	Male	73.3	46.7	5.8	1.8	100.0	50.0	13.4	2.2
S6	Male	70.0	43.3	5.2	1.5	80.3	50.0	7.3	2.2
S7	Male	93.3	56.7	10.7	3.1	83.0	33.3	7.9	0.6
S8	Male	70.0	46.7	5.2	1.8	76.0	36.7	6.4	0.8
S9	Female	80.0	36.7	7.2	0.9	73.0	40.0	5.8	1.1
S10	Male	86.7	46.7	8.8	1.8	76.6	46.0	6.5	1.7
Mean ± SD	-	81.3 ± 9.8	46.4 ± 6.4 *	7.8 ± 2.6	1.8 ± 0.7 *	83.4 ± 8.5	43.6 ± 7.9 *	8.3 ± 2.3	1.6 ± 0.8 *

Note: * *p* value < 0.001 with paired *t* tests.

**Table 2 brainsci-12-01152-t002:** Accuracy and ITR with the SSVEP-based BCI.

Subjects	Sex	Accuracy (%) by CCA Decoding		ITR (Bits/min) by CCA Decoding		Accuracy (%) by FBCCA Decoding		ITR (Bits/min) by FBCCA Decoding	
		Normal	LME	Normal	LME	Normal	LME	Normal	LME
S1	Female	83.3	56.7	18.6	7.1	90.0	53.0	22.6	6.0
S2	Female	93.3	50.0	24.9	5.1	90.6	52.3	23.0	5.7
S3	Male	93.3	53.3	24.9	6.0	93.0	57.7	24.7	7.4
S4	Female	86.7	36.7	20.5	2.0	100.0	66.3	31.0	10. 6
S5	Male	90.0	40.0	22.6	2.7	90.3	52.7	22.8	5.9
S6	Male	86.7	36.7	20.5	2.0	100.0	40.3	31.0	2.7
S7	Male	93.3	56.7	24.9	7.1	86.7	49.7	20.5	5.0
S8	Male	86.7	33.3	20.5	1.4	95.0	50.0	26.2	5.1
S9	Female	96.7	60.0	27.6	8.2	93.0	66.7	24.7	10.7
S10	Male	100.0	63.3	31.0	9.4	94.7	49.0	26.0	4.8
Mean ± SD	-	91.0 ± 5.2	48.7 ± 11.0 *	23.6 ± 3.8	5.1 ± 2.9 *	93.3 ± 4.3	53.7 ± 8.0 *	25.2 ± 3.5	6.4 ± 2.5 *

Note: * *p* value < 0.001 with paired *t* tests.

**Table 3 brainsci-12-01152-t003:** Overall Scores from the NASA-TLX.

Subjects	Mental Demand		Physical Demand		Temporal Demands		Frustration		Effort		Performance	
	Normal	LME	Normal	LME	Normal	LME	Normal	LME	Normal	LME	Normal	LME
S1	50	85	50	85	40	90	45	95	50	75	50	85
S2	50	90	55	90	60	80	35	100	40	90	45	95
S3	35	95	35	90	45	85	25	95	50	90	45	90
S4	35	90	40	95	45	80	40	90	40	90	40	95
S5	45	80	45	85	35	85	50	85	55	90	45	80
S6	35	90	35	100	30	95	25	95	50	85	30	90
S7	35	80	25	80	35	90	40	75	35	75	35	80
S8	25	85	15	80	15	90	20	75	30	75	15	90
S9	15	70	5	60	5	60	10	55	15	50	5	70
S10	15	70	25	60	10	80	20	75	25	75	15	70
Mean ± SD	34.0 ± 12.6 *	83.5 ± 8.5	33.0 ± 15.7 *	82.5 ± 13.4	32.0 ± 17.4 *	83.5 ± 9.7	31.0 ± 13.0 *	84.0 ± 14.0	39.0 ± 13.0 *	80.0 ± 12.6	32.5 ± 15.7 *	84.5 ± 9.3

Note: * *p* value < 0.001 with paired *t* tests.

**Table 4 brainsci-12-01152-t004:** Overall Scores from the Fatigue Questionnaire.

Subjects	Physical Fatigue		Mental Fatigue	
	Normal	LME	Normal	LME
S1	13	88	34	67
S2	25	88	34	67
S3	13	88	17	50
S4	13	100	50	50
S5	45	100	50	67
S6	25	75/	34	67
S7	20	75	34	67
S8	25	88	17	67
S9	15	75	17	67
S10	10	75	34	50
Mean ± SD	20.4 ± 10.4 *	85.2 ± 9.9	32.1 ± 12.2 *	62.0 ± 8.2

Note: * *p* value < 0.001 with paired *t* tests.

## Data Availability

All data generated or analyzed during this study can be obtained by emailing the corresponding author.

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
