# Peer review of "Effects of Low Mental Energy from Long Periods of Work on Brain-Computer Interfaces"

_brainsci, 2022, doi:10.3390/brainsci12091152_

Round 1

Reviewer 2 Report

·      Page 2 Line 86 – “the subjects were asked to work continuously for a long time”

o   What is in scope for “work”?

o   What is considered “a long time”? 

o   How were work and time assessed, reported, and measured?

§  If these details are described entirely later on under “experimental procedure,” please refer to that section in the paragraph about subjects – or delete the line about what subjects were asked to do under subjects.

·      Authors cite Murre et al. Plos One 2015 for the claim that “subjects would forget most experimental details” within a 3-day interval.  However based on my interpretation of the Ebbinghaus Forgetting Curve reported in Murre 2015, it appears that subjects retain 50% or more information over a 3-day period. How do you define “most experimental details”? Please quantify with an evidence-based percentage of experimental details.

·      Page 6, line 221: “each 700-ms-long EEG signal was down-sampled to a duration of 70 ms.” – do the authors mean that the 700 ms long EEG signal sampled at 200 Hz was downsampled to a 700 ms long EEG signal sampled at 20 Hz? If a “70 ms” window covers 700 ms of time, it’s still a 700 ms window!

·      It’s not clear to the reviewer that the frequency bands on the bubble sort algorithm are compatible with the downsampled data.  A 41-46 Hz frequency band (SB_7) requires data sampled at > 92 Hz to be resolved. However the sampling rate of the “70 ms” window is not clear from the information given. Please clearly explain how the frequency bands listed on line 258 can be resolved at Nyquist with the data processing algorithm as implemented here.

·      Is normal condition always recorded earlier in the day? Is LME condition always recorded 18 hours later? Without a control normal condition after 18 hours of wakefulness, how can the authors attribute changes in neural signal to LME and not just circadian/wake effects? How can you differentiate between impacts of long periods of work and long periods of wakefulness? If this was explained in the paper, please outline it more clearly.

Round 2
